# Dog Owners’ Perceptions of the Convenience and Value of Chewable Oclacitinib: Quantitative Survey Data from an International Survey

**DOI:** 10.3390/ani14060952

**Published:** 2024-03-19

**Authors:** Andrea Wright, Andrew Hillier, Jonathan Lambert, Kennedy Mwacalimba, Natalie Lloyd, Tetsushi Kagiwada, Yoriko Hashiguchi, Carolyne Hours, Danielle Riley, Ashley Enstone, Robin Wyn

**Affiliations:** 1Zoetis US, Parsippany, NJ 07054, USA; 2Zoetis UK, Leatherhead, Surrey KT22 7LP, UK; 3Zoetis NZ, Auckland 1010, New Zealand; 4Zoetis Japan, Shibuya-Ku, Tokyo 151-0053, Japan; 5Zoetis Canada, Kirkland, QC H9H 4M7, Canada; carolyne.hours@zoetis.com; 6Adelphi Values PROVE, Bollington, Cheshire SK10 5JB, UK

**Keywords:** oclacitinib, canine dermatitis, canine pruritus, pet owner preference, treatment convenience, treatment compliance

## Abstract

**Simple Summary:**

Oclacitinib is a tablet-based therapy given for itching that is caused by allergic or atopic dermatitis in dogs. The objective of this study was to understand pet owners’ perceptions of the conventional film-coated form of oclacitinib and the chewable form of oclacitinib in terms of convenience and value. Firstly, an interview phase with pet owners and veterinarians was used to develop detailed written profiles of these treatment options. Then, pet owners were invited to participate in a survey about their experiences and preferences. Overall, 1590 pet owners provided survey responses. Most respondents (62%) reported having experienced challenges giving tablet-based therapies to their dog(s), and half of the respondents (52%) had experience giving flavoured or chewable tablets to their dog. Comparing oclacitinib and chewable oclacitinib (with or without associated costs), the majority of the respondents preferred the chewable formulation in all countries for both short-term and long-term itch (≥58%; all *p* < 0.05). Chewable and/or palatable treatment options may be welcomed by pet owners and may have potential positive impacts on convenience, compliance, outcomes, quality of life, and the human–animal bond.

**Abstract:**

Oclacitinib is an oral therapy indicated for pruritus associated with allergic or atopic dermatitis in dogs. This study sought to assess pet owners’ perceptions of the relative convenience and value of the conventional film-coated formulation and the chewable formulation. A quantitative discrete-choice experimental methodology was applied, comparing (conventional, film-coated) oclacitinib versus chewable oclacitinib using unbranded treatment profiles. Initially, a qualitative interview phase with pet owners and veterinarians was conducted to develop detailed treatment profiles. Subsequently, pet owners participated in a quantitative survey. Overall, 1590 pet owners provided survey responses. Most respondents (62%) reported having experienced challenges administering tablet-based therapies to their dog(s). Half of all respondents (52%) had experience administering flavoured or chewable tablets to their dog. Comparing oclacitinib and chewable oclacitinib (with or without associated costs), the majority of the respondents preferred the chewable formulation in all regions across short-term and long-term scenarios (≥58%; all *p* < 0.05). The current research is one of few survey-driven studies for treatment preferences in companion animal medicine. Veterinarians may offer chewable or palatable treatment options where available, with potential positive impacts on convenience, compliance, outcomes, quality of life, and the human–animal bond.

## 1. Introduction

Allergic dermatitis is among the most commonly observed medical conditions in pet dogs [1], and pruritus (itching) is the hallmark clinical sign [2]. Pruritic behaviours are likely to impact the quality of life (QoL) of the dog and owner [3] and may contribute to the owner’s decision to seek pharmacological treatment for their dog.

There appears to be increasing interest in ensuring the comfort of pet owners and their companion animals, including aspects such as nutrition and medication. Previous research has examined potential health hazards resulting from feeding practices [4].

Several pharmacological treatment options are available for the management of canine pruritus [5], including corticosteroids, cyclosporine, allergen immunotherapy, antihistamines, and newer therapies such as oclacitinib and lokivetmab [6,7]. In addition, these therapies may also be available in multiple formulations for oral administration, whose characteristics may influence the likelihood of successfully and easily administering the therapy to a pet dog. Such characteristics may include the smell, taste, texture, size, and appearance of a tablet, each contributing to its overall palatability (which can be defined as the likelihood that a tablet will be spontaneously consumed within a certain timeframe) [8,9].

Low palatability may contribute to frustration and/or stress for the pet and owner, reduced compliance, and worsened outcomes [10]. Palatable treatments may not need to be administered in food and therefore may alleviate potential microbiological contamination concerns [4]. Therefore, improving palatability of a medication may subsequently lead to improved outcomes and therefore an improved pet–owner bond.

Oclacitinib is an oral therapy indicated for the control of pruritus associated with allergic or atopic dermatitis in dogs at least 12 months of age [11,12]. Currently, oclacitinib is available in both a conventional film-coated formulation and a chewable formulation with confirmed high palatability [13]. However, pet owners’ perceptions of the relative convenience and value of these two options are not known.

In light of the above, a quantitative discrete-choice experimental study methodology was applied to determine pet owners’ preferences for (conventional, film-coated) oclacitinib versus chewable oclacitinib for canine pruritus, using unbranded treatment profiles. In this experiment type, respondents are instructed to select their preferred option from a set of two or more alternatives, through making trade-offs between the specific desirable and/or undesirable characteristics of each. In the current study, treatment profiles were unbranded and focused on the physical characteristics of each treatment option, in order to avoid potential bias. Qualitative research was undertaken prior to development of a quantitative survey, and this survey was localized and undertaken by pet owners in five regions (Canada, France, Japan, New Zealand, and the United Kingdom).

## 2. Materials and Methods

Initially, a qualitative interview phase with pet owners and veterinarians (*n* = 12) was conducted to allow the development of detailed profiles of oclacitinib and chewable oclacitinib, for incorporation into a qualitative survey. Subsequently, a quantitative survey was designed and launched for completion in a sample of pet owners (*n* = 1590).

### 2.1. Qualitative Phase (Survey Development)

A qualitative interview phase was conducted to develop detailed profiles of oclacitinib and chewable oclacitinib, for incorporation into a qualitative survey.

Interviews were initially conducted with pet owners in the UK (*n* = 4), in order to understand the nature of difficulties that may be experienced when administering conventional film-coated-tablet-based therapies to pet dogs. Each respondent was an owner of at least one dog diagnosed with pruritus and who had experienced difficulties with administering treatment. These interviews included exploratory questions pertaining to each pet owner’s experience with their current treatment regimen and perceptions of conventional and chewable tablet/capsule-based therapies. Probing questions were utilized exploring the process of administration, convenience, and their dog’s response to taking their current treatment, including any challenges experienced with conventional tablet or capsule-based therapies, any key differences identified when considering conventional versus chewable-based therapies, and key factors of importance when considering the administration of conventional versus chewable therapies. Pet owners were also invited to rate the importance of several factors that may be affected by treatment difficulties. 

These initial interviews with pet owners identified several key considerations that were influenced by difficulties with administering treatment. Factors that were highly ranked, in terms of importance, by pet owners included the following: disruption to the pet–owner bond and lifestyle, feeling inadequate due to uncertainty as to whether your pet’s condition is being managed appropriately, worry relating to the effectiveness of the treatment due to missed doses, and reduced ability to allow others to look after your pet. Insights derived from these pet owner interviews were then used to develop draft profiles of oclacitinib and chewable oclacitinib. In addition, clinical data on the palatability of the chewable formulation were used to inform the wording [13].

Further qualitative interviews were subsequently conducted with veterinarians in Japan, New Zealand, the UK, and the US (*n* = 2 in each region; *n* = 8 in total). These interviews again explored difficulties that are reported by pet owners to veterinarians when administering conventional (film-coated) tablet-based therapies and chewable tablet-based therapies. These interviews included exploratory questions pertaining to the management of canine pruritus with tablet- or capsule-based therapies, as well as key features of such therapies, the influence of these features on the selection process, and characteristics of an ideal therapy, for canine pruritus and other therapy areas.

In addition, these veterinarians reviewed the draft treatment profiles and proposed changes in order to ensure clinical accuracy and comprehensibility (from a pet owner’s perspective).

These validation interviews with veterinarians identified several further themes relating to difficulties with administering treatment. These themes, along with direct comments and suggestions for improvement made on the draft treatment profiles, were used to develop final treatment profiles for use in the quantitative survey.

### 2.2. Quantitative Phase (Survey Conduct)

A quantitative survey was designed and launched for completion in a sample of pet owners from Canada, France, Japan, New Zealand, the UK, and the US, with a target sample size of *n* = 250 per region (*n* = 1500 total), consistent with previous preference research studies seen in the published literature [14,15,16,17,18,19,20,21,22,23,24,25].

This survey was designed to determine pet owners’ willingness to pay (WTP) for a film-coated pill-based therapy versus a chewable formulation for the treatment of canine pruritus, through presentation of detailed product profiles that respondents were instructed to evaluate and select between (see Appendix A for details of the tested treatment profiles).

Pet owners (owning at least one dog; with or without previous experience with pruritus) were invited to participate in the survey. Other criteria for inclusion in the survey were as follows: being aged ≥18 and resident in one of the regions of interest; owning no more than three dogs and cats in total; being the primary caretaker/decision maker for their dog(s); and not being employed in animal health or market research. In addition, quotas were applied for residence by sub-region within each region, to ensure that a geographically representative sample was collected. Recruited individuals were pre-registered within research panels maintained by a research agency (Qualtrics, Seattle, WA, USA) and were contacted by email for potential participation.

The quantitative survey (see Figure 1) primarily employed a discrete-choice experimental design, where respondents selected from two treatment profiles (by making trade-offs between the specific characteristics of those profiles). Use of this methodology allowed oclacitinib and chewable oclacitinib profiles to be compared directly and also allowed for a variety of specific cost levels to be examined. See Appendix A for details of the tested treatment profiles.

Two types of choice questions were included. One type of question asked whether pet owners would be willing to pay an additional cost to use chewable oclacitinib rather than oclacitinib. From this question type, differences in preference between oclacitinib and chewable oclacitinib treatment profiles were later examined using chi-squared analysis. The second question type asked whether pet owners would prefer oclacitinib or chewable oclacitinib, either in the absence of any associated cost or in a scenario where chewable oclacitinib was 5% cheaper, equivalently priced, or 5% more expensive than oclacitinib. From this question type, regression analyses were conducted to examine the influence of subgroup characteristics on the likelihood of being willing to pay an additional cost to use chewable oclacitinib rather than oclacitinib.

Other question designs including rating and ranking tasks were also included. Broadly, this survey initially established pet owners’ eligibility and willingness to participate, before exploring their experience with administration difficulties and then their preferences for oclacitinib versus chewable oclacitinib.

Realistic “out of pocket” costs for oclacitinib and chewable oclacitinib were developed prior to the roll-out of the survey and were informed by the expertise of regional representatives of the sponsoring company. Individual costs were developed specifically for each region (incorporating drug cost, retailer markup, and tax), and also took into account that some therapy options are dosed variably based on weight. Specifically, a weight of 27 kg, representing a medium-sized pet dog (e.g., adult Labrador), was used in all regions except Japan, where a smaller standard weight of dog was assumed (7.5 kg), due to differing trends in pet demographics.

Data were collected for two hypothetical treatment scenarios: a short-term pruritus scenario that was defined as: “itch which will resolve itself within 14 days, but it should be treated in the meantime”; a long-term pruritus scenario that was defined as: “it is uncertain when and if this condition will resolve itself”.

Prior to full roll-out of the survey in all regions, a pilot sample of 25 responses from UK pet owners was collected in order to confirm the overall comprehensibility of the survey, prior to localization for each individual region.

Following collection of the target sample size of responses in each region, survey results were collated and underwent quality checks prior to analysis (including removal of highly unrealistic suggested WTP values or incoherent free-text responses).

## 3. Results

### 3.1. Characteristics of Survey Respondents

A total of 1590 pet owners provided survey responses, fulfilling the target sample size in all regions (including *n* = 264 Canada, *n* = 271 France, *n* = 267 Japan, *n* = 264 New Zealand, and *n* = 262 UK, and *n* = 262 US respondents).

The demographic details (across regions) are presented in Table 1, while the details of pet ownership (including insurance status) are presented in Table 2, and the details of experiences with canine pruritus are presented in Table 3. Of note, 42% of the respondents held pet insurance for their dog, and 54% had current or past experience of canine pruritus.

Pet owners’ experiences and perceived challenges with tablet-based treatments were ascertained, in order to test if these may influence WTP for chewable options. The majority of the respondents (978 of 1590; 62%) reported having experienced challenges administering tablet-based therapies to their dog(s). In addition, approximately half of these respondents (515 of 978; 32% of all respondents) reported challenges even when tablets had been placed in food (see Table 4). The majority of these respondents (628 of 978) reported that such challenges were experienced for one month (or a shorter period of time; see Table 5).

Table 6 presents challenges and required adaptations reported by survey respondents in the context of administering tablet-based therapies to their pet dog(s), among the subgroup of 978 pet owners who reported these problems. A total of 556 (56.9%) out of those 978 respondents reported “very often” or “almost constantly” experiencing two or more of these challenges, and 396 (40.5%) out of 978 reported “very often” or “almost constantly” experiencing three or more. Of note, 59% of 978 respondents reported that they often had to place tablets in food to ensure that these were consumed.

Half of all survey respondents (52%) had experience administering flavoured and/or chewable tablets to their dog, while 6% had experience with both types of formulation (see Table 7). Among those with experience administering flavoured and/or chewable tablets (*n* = 830), respondents most commonly had a neutral (32%), positive (43%), or very positive (17%) perception of these formulations (see Table 8).

### 3.2. Treatment Preferences

All respondents (*n* = 1590) were asked for their preferences relating to unbranded treatment profiles.

Comparing product profiles representing oclacitinib and chewable oclacitinib (without associated costs), the majority of all respondents stated a preference for the chewable formulation in all regions (≥75% in Canada; ≥68% in France; ≥75% in Japan; ≥77% in New Zealand; ≥76% in the UK; ≥64% in the US), across both short-term and long-term pruritus scenarios (see Figure 2; all *p* < 0.05).

When hypothetical out-of-pocket costs were added (pricing chewable oclacitinib 5% lower, equally to, and 5% higher than the cost of oclacitinib, respectively), the majority of the respondents again stated a preference for the chewable formulation in all regions (≥71% in Canada; ≥65% in France; ≥65% in Japan; ≥64% in New Zealand; ≥69% in the UK; ≥58% in the US), across both short-term and long-term pruritus scenarios (see Figure 3; all *p* < 0.05).

### 3.3. Willingness to Pay (WTP) for Chewable Oclacitinib

All respondents (*n* = 1590) were asked whether they would be willing to pay an additional cost to receive chewable oclacitinib rather than oclacitinib.

When asked to state their WTP an additional cost to receive chewable oclacitinib rather than oclacitinib, approximately one-third of the respondents were willing to pay (≥31% in Canada; ≥30% in France; ≥39% in Japan; ≥32% in New Zealand; ≥32% in the UK; ≥31% in the US), across both short-term and long-term pruritus scenarios (see Table 9).

In regression analyses, several subgroup characteristics were found to be significantly associated with increased or decreased WTP an extra cost for chewable oclacitinib:Japanese respondents were more likely to be willing to pay in the acute scenario: coefficient +0.134 (standard error ±0.049) (*p* < 0.01).UK respondents were less likely to be willing to pay in the chronic scenario: −0.327 (±0.050) (*p* < 0.01).Respondents from the highest income category were more likely to be willing to pay in both acute and chronic scenarios: +0.128 (±0.050) and +0.123 (±0.048), respectively (both *p* < 0.05).Respondents working part time were more likely to be willing to pay in the chronic scenario: +0.100 (±0.049) (*p* < 0.05).Respondents with experience of challenges with administering tablet-based therapies even when the tablet is placed in food were more likely to be willing to pay in both acute and chronic scenarios: +0.062 (±0.030) and +0.069 (±0.029), respectively (both *p* < 0.05).

## 4. Discussion

This is a two-phase study that collected quantitative data from 1590 pet owners across five different countries. The initial qualitative phase involved the development of detailed profiles of two formulations of oclacitinib and the identification of challenges related to administering tablet-based treatments and was followed by the quantitative phase, which estimated respondents’ WTP for chewable oclacitinib versus oclacitinib.

Overall, 62% of all respondents reported experiencing challenges while administering tablet-based therapies to their pet dogs, with 37% of those respondents reporting that they very often or almost constantly hide the tablet-based medication within food for their pet to ingest. Some pet owners also were worried that their pet was not receiving the full benefit of treatment due to administration issues.

Overall, 52% of the survey respondents had some experience with administering flavoured or chewable formulations to their pet dogs, with 60% of these pet owners having a positive reaction towards these options.

When asked if they were willing to pay more to use chewable oclacitinib, between 30 and 42% of the respondents were willing to pay, dependent on the region and pruritus scenario. However, in specific long-term and short-term canine pruritus scenarios, chewable oclacitinib was favoured by pet owners, even with an associated hypothetical cost 5% greater than that of oclacitinib. 

The current research is one of few survey-driven studies looking at pet owner preferences for treatment options in companion animal medicine. The sample size collected by this study (*n* ≥ 262 pet owners per region) is consistent with previous preference research in human and animal health, which typically involve 100 to 500 respondents [14,26,27,28].

There are some limitations to consider with the current methodology. The first arises from the potential of participation bias. This study enrolled pet owners who had previously expressed interest in participating in online survey-based research and who were willing to participate in the current study. Therefore, there is a risk that the study sample is skewed towards certain demographics who are more likely to participate, rather than being representative of pet owners as a whole in each region. However, detailed demographic information has been collected and reported here, to allow any potential participation bias to be identified. 

In addition, WTP for chewable oclacitinib at a higher cost than oclacitinib may have varied according to the socioeconomic situation in each region at the time of the survey.

Out of the 1590 respondents, only 863 have had previous experience with canine pruritis. The responses from the remaining pet owners, with no prior experience in managing canine pruritis, were therefore likely influenced primarily by the provided descriptions of the two oclacitinib formulations. However, extensive qualitative research was undertaken prior to the survey in order to ensure that the provided descriptions of the pruritus scenarios and treatments were representative and comprehensible.

## 5. Conclusions

This study demonstrated that chewable oclacitinib was favoured by pet owners in comparison to oclacitinib for treating canine pruritis, with up to 42% of the respondents willing to pay more for chewable oclacitinib. Therefore, veterinarians may be able to offer chewable and/or palatable treatment options where available, with potential subsequent positive impacts on convenience, compliance, outcomes, quality of life, and subsequently the human–animal bond. Future research in this area could establish whether prior experience with administering chewable and/or flavoured canine medications influences the level of preference that is stated for alternative formulations.

## Figures and Tables

**Figure 1 animals-14-00952-f001:**
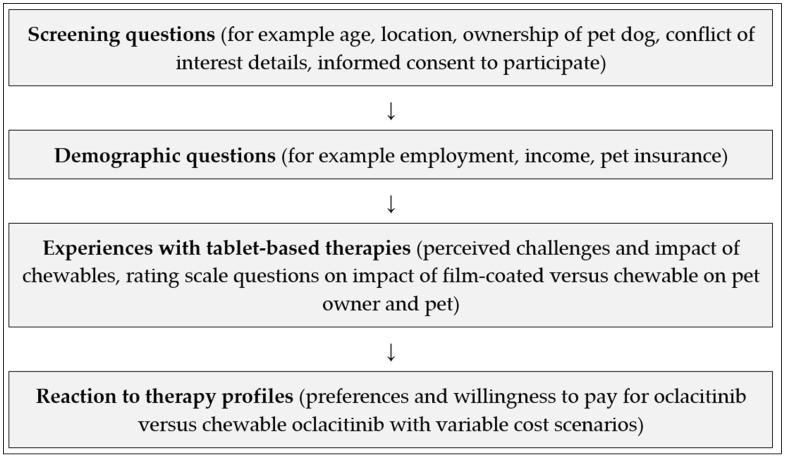
Overview of structure of quantitative pet owner survey.

**Figure 2 animals-14-00952-f002:**
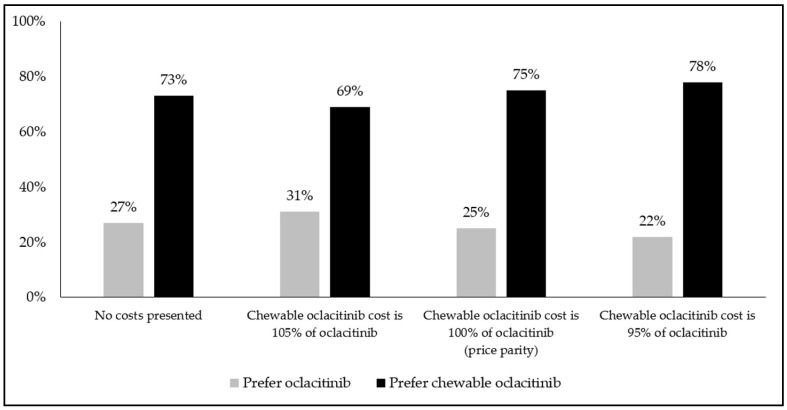
Respondents’ preferences for oclacitinib or chewable oclacitinib product profiles, with or without associated costs, for short-term pruritus (*n* = 1590); short-term pruritus scenario was defined as: “itch which will resolve itself within 14 days, but it should be treated in the meantime”; all comparisons *p* < 0.05.

**Figure 3 animals-14-00952-f003:**
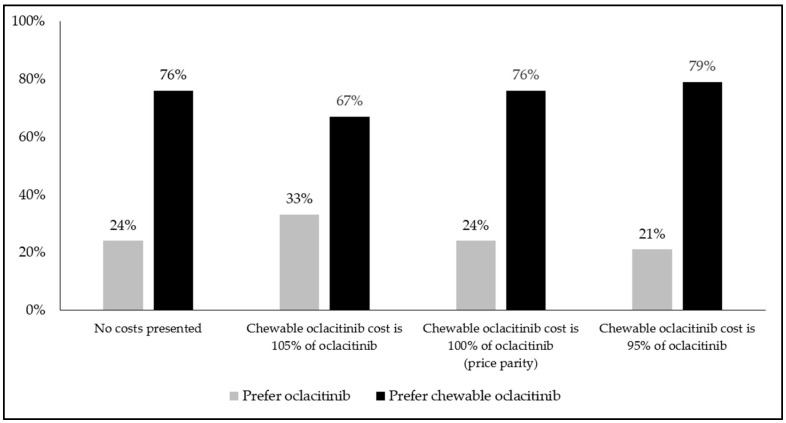
Respondents’ preferences for oclacitinib or chewable oclacitinib product profiles, with or without associated costs, for long-term pruritus (*n* = 1590); long-term pruritus scenario was defined as: “it is uncertain when and if this condition will resolve itself”; all comparisons *p* < 0.05.

**Table 1 animals-14-00952-t001:** Survey respondents’ (*n* = 1590) demographic details.

Age	*n*	%
18–30	258	16%
31–40	356	22%
41–50	319	20%
51–60	265	17%
>60	392	25%
Gender	*n*	%
Female	861	54%
Male	725	46%
Gender-diverse	3	0%
Prefer not to say	1	0%
Education	*n*	%
Primary school	17	1%
Secondary/high school	385	24%
Diploma or vocational qualification	378	24%
Undergraduate degree	548	34%
Postgraduate degree	235	15%
Doctorate	27	2%
Work status	*n*	%
Full-time worker	920	58%
Part-time worker	200	13%
Retired	268	17%
Student	26	2%
Unemployed	176	11%
Household income *	*n*	%
First (lowest) income band	290	18%
Second income band	578	36%
Third (middle) income band	315	20%
Fourth income band	192	12%
Fifth (highest) income band	215	14%

* Five income bands were defined for each region, based on local currency and average income levels.

**Table 2 animals-14-00952-t002:** Survey respondents’ (*n* = 1590) pet ownership details.

Dogs in Household	*n*	%
1	1383	87%
2	190	12%
3	17	1%
Cats in household	*n*	%
0	931	59%
1	541	34%
2	118	7%
Total pets in household (dogs + cats)	*n*	%
1	792	50%
2	595	37%
3	203	13%
Insurance	*n*	%
Yes, I hold pet insurance for my dog	666	42%
No, I do not hold pet insurance for my dog	924	58%
Average amount (%) paid for medical costs for this dog (from own pocket), when claiming medical costs using this insurance (among *n* = 572 of *n* = 666 who knew this; *n* = 94 did not know this amount)	30.65% (IQR: 15.00% to 40.00%)	9%

**Table 3 animals-14-00952-t003:** Survey respondents’ (*n* = 1590) experiences with canine pruritus and its treatment.

Current or Past Owner of a Dog That Has Experienced Pruritus	*n*	%
Yes	863	54%
No	727	46%
Current owner of a dog that has experienced pruritus	n	%
Yes	617	39%
No	973	61%
Current dog has pruritus that was diagnosed <12 months ago	n	%
Yes	173	28%
No	435	72%
Current dog is receiving or has received pharmacological medication for pruritus **	n	%
Yes	538	87%
No	36	6%
Do not know	43	7%
Previously owner of a dog that experienced pruritus	*n*	%
Yes	636	40%
No	954	60%
Previous dog received pharmacological medication for pruritus *	*n*	%
Yes	557	87%
No	23	4%
Do not know	56	9%

* Five income bands were defined for each region, based on local currency and average income levels. ** Respondents selected from a list that included prescription therapies (e.g., steroids, oclacitinib, cyclosporine, lokivetmab) and over-the-counter therapies (e.g., antihistamines).

**Table 4 animals-14-00952-t004:** Respondents (*n* = 1590) who had experienced or not experienced challenges administering tablet-based therapies to their dog(s).

	Have Challenges Been Experienced When Administering Tablet-Based Therapies to Your Dog(s)	
Yes	Yes, this problem has occurred, whether or not tablets were given directly or placed in food	515 (32%)
	Yes, this problem has occurred, but never when tablets are placed in food	463 (29%)
No	No, this problem has never occurred	612 (38%)

**Table 5 animals-14-00952-t005:** Length of time that challenges with administration of tablet-based therapies have been experienced by respondents (*n* = 978).

Length of Time That Challenges Have Been Experienced	*n*	%
1 month	628	64%
6 months	183	19%
12 months	35	4%
>12 months	132	13%

**Table 6 animals-14-00952-t006:** Challenges reported by respondents (*n* = 978) related to administering tablet-based therapies to their dog(s).

“How Often Have You Experienced the Following Issues When Administering Tablet-Based Treatments to Your Dog(s)? (With 1 Being Almost Never and 7 Being Almost Constantly)”			
Issue	Mean Score	% Answering 1 or 2	% Answering 6 or 7
Administration issues are negatively influencing your relationship with your dog.	3.03	48%	12%
You are worried that your dog is not properly experiencing the benefit of the tablet.	**4.11**	**24%**	**24%**
You feel stressed or unhappy about the difficulties you are experiencing with the tablet.	3.85	28%	20%
You feel that your dog is wary of you or otherwise less likely to interact with you as they normally would.	3.57	35%	18%
You have difficulties remembering or organizing to administer the tablet, due to your dog’s reluctance to take this.	3.06	48%	14%
You must break up the tablet and place it into food (for example, by snapping or grinding up the tablet; otherwise, your dog will not take the tablet).	**4.50**	**21%**	**37%**
You must build extra time into your day or interrupt your daily routine, in order to administer the tablet.	3.65	35%	22%
You must physically administer the tablet to your dog (for example, by opening their mouth and putting the tablet on the back of the tongue and confirming they have swallowed the tablet).	3.83	32%	**24%**
You must place the tablet into a piece of food or cover the tablet with food (otherwise, your dog will not take the tablet).	**5.59**	**5%**	**59%**
Your dog reacts badly to you trying to administer the tablet (for example, by barking or biting you).	3.00	49%	13%
Your dog runs away or hides from you, when it becomes aware that you are trying to administer the tablet.	3.61	35%	19%
Your dog spits out their tablet or otherwise leaves it somewhere without consuming it; you are concerned about whether or not your dog has taken their medicine.	**4.03**	**27%**	**25%**

Note: bolded and shaded results indicate the most common issues (i.e., high average score, high proportion answering 6 or 7; low proportion answering 1 or 2).

**Table 7 animals-14-00952-t007:** Respondents’ (*n* = 1590) level of experience with administering flavoured and/or chewable tablets.

Experience with Administering Flavoured and/or Chewable Tablets	*n*	%
Yes		
Yes, total	830	52%
Yes, flavoured tablets + chewable tablets	96	6%
Yes, flavoured tablets only	457	29%
Yes, chewable tablets only	277	17%
No		
No	760	48%

**Table 8 animals-14-00952-t008:** Perception of flavoured and/or chewable tablets among respondents who had experience administering these (*n* = 830).

Perception of Flavoured and/or Chewable Tablets	*n*	%
Very negative	13	2%
Negative	51	6%
Neutral	267	32%
Positive	355	43%
Very positive	144	17%

**Table 9 animals-14-00952-t009:** Respondents’ willingness to pay (WTP) an additional cost * for a chewable formulation of oclacitinib for short-term and long-term pruritus.

WTP for Short-Term Pruritus in Canada (*n* = 264)		
Yes	31%	Mean (range): CAD 99.06 (5.0 to 1000.0)
No	69%	--
WTP for Long-Term Pruritus in Canada (*n* = 264)		
Yes	34%	Mean (range): CAD 86.64 (1.3 to 500.0)
No	66%	--
WTP for Short-Term Pruritus in France (*n* = 271)		
Yes	30%	Mean (range): EUR 56.99 (1.0 to 200.0)
No	70%	--
WTP for Long-Term Pruritus in France (*n* = 271)		
Yes	32%	Mean (range): EUR 56.10 (2.0 to 160.0)
No	68%	--
WTP for Short-Term Pruritus in Japan (*n* = 267)		
Yes	42%	Mean (range): JPY 3391.68 (1.0 to 15,000.0)
No	58%	--
WTP in Long-Term Pruritus in Japan (*n* = 267)		
Yes	39%	Mean (range): JPY 3119.82 (1.0 to 90,000.0)
No	61%	--
WTP for Short-Term Pruritus in NZ (*n* = 264)		
Yes	32%	Mean (range): NZD 137.19 (5.0 to 1000.0)
No	68%	--
WTP for Long-Term Pruritus in NZ (*n* = 264)		
Yes	33%	Mean (range): NZD 136.49 (5.0 to 1000.0)
No	67%	--
WTP for Short-Term Pruritus in the UK (*n* = 262)		
Yes	32%	Mean (range): GBP 62.21 (2.0 to 200.0)
No	68%	--
WTP for Long-Term Pruritus in the UK (*n* = 262)		
Yes	34%	Mean (range): GBP 73.99 (3.0 to 200.0)
No	66%	--
WTP for Short-Term Pruritus in the US (*n* = 262)		
Yes	32%	Mean (range): USD 59.71 (1.0 to 150.0)
No	68%	--
WTP for Long-Term Pruritus in the US (*n* = 262)		
Yes	31%	Mean (range): USD 65.61 (5.0 to 150.0)
No	69%	--

* Treatment for short-term pruritus was described as 28 tablets over 14 days; specified costs for (conventional, film-coated) oclacitinib in this scenario were: 116.26 CAD; 125.83 EUR; 8974.35 JPY; 161.68 NZD; 97.73 GBP; and 85.84 USD per course. Treatment for long-term pruritus was described as 30 tablets over 30 days, repeated monthly; specified costs for (conventional, film-coated) oclacitinib in this scenario were: 123.14 CAD; 132.67 EUR; 9615.38 JPY; 171.45 NZD; 104.28 GBP; and 90.90 USD per month. WTP: willingness to pay.

## Data Availability

The datasets used and/or analysed during the current study are available from the corresponding author on reasonable request.

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
