# Peer review of "Dog Owners’ Perceptions of the Convenience and Value of Chewable Oclacitinib: Quantitative Survey Data from an International Survey"

_animals, 2024, doi:10.3390/ani14060952_

Round 1

Reviewer 1 Report

Comments and Suggestions for Authors

Thank you for sharing the results of your study into the response of dog owners to the two administration formats of Oclacitinib.  The scientific method is well described and generally well executed.  The findings are not entirely surprising, given the well known issues associated with regular dosing of dogs with traditional orally administered medication.

I would make the following comments about the manuscript:

Line 24; when referring to the 30%, why is this number quoted as the 'headline' when surely the more enlightening figure would be the approximately 70% that were not prepared to pay more?

Section 3.1; Qualitative Phase - how exactly were these interviews used to inform the qualitative phase.  Were all of the details used, or was the information used selectively? If the latter, based on what criteria, most commonly given answers, or responses most relevant to your research question?  Were the vet opinions used to endorse the Pet Owners responses as well as give their own input?  It is important to know exactly how the research questions were derived.

Line 162 - Can you define the WTP abbreviation prior to this point.

Line 162 - what constitutes an 'unrealistic WTP?  In table 9 there are already a wide range of values quoted.

What are the 'Don't Knows' on line 182.  This brings the total to 1733.

Table 6; why was a 7 point Likert scale used? Is this a scale which you have validated for the question types?  What were the other descriptors used for intervening points?

Table 7; I think it would have been valuable to interrogate more the data on the 96 respondents that had previous experience of both formats.  Admittedly, it is not a large group, but their experience would be quite informative nonetheless.

There appears to be some discrepancy in the attitudes of respondents illustrated between Figures 2/3 and the WIP results of Table 9.  Although the preference for chewable format remained high in table 2 and 3 even with the increased cost, the majority of people clearly are not prepared to pay more as illustrated in Table 9.  This may be worth discussing I think.

Reviewer 2 Report

Comments and Suggestions for Authors

The paper, titled "Dog owners’ perceptions of the convenience and value of chewable oclacitinib: quantitative survey data from an international survey," aims to assess pet owners' perceptions of the convenience and value of the conventional film-coated formulation and the chewable formulation of oclacitinib, an oral therapy for itching caused by allergic or atopic dermatitis in dogs. The study used a quantitative discrete-choice experiment methodology to compare the two formulations and found that the majority of respondents preferred the chewable formulation in both short-term and long-term itch scenarios. Additionally, a significant percentage of respondents were willing to pay more for the chewable oclacitinib. The paper's main contributions lie in providing insights into pet owners' preferences for different treatment options for canine pruritus, which can have potential positive impacts on convenience, compliance, outcomes, quality of life, and the human-animal bond.

The paper falls within the scope of the journal as it addresses a topic related to companion animal medicine, specifically focusing on pet owners' preferences for different treatment options for canine pruritus.

General concept comments:

The paper's main question is addressed by the research, and the topic is relevant in the field as it provides valuable insights into pet owners' preferences for different treatment options for canine pruritus.

The paper adds to the subject area by conducting a quantitative survey to assess pet owners' perceptions of the convenience and value of the conventional film-coated formulation and the chewable formulation of oclacitinib, which is a relatively novel approach in companion animal medicine.

The authors should consider providing more details about the methodology, such as the specific design of the discrete-choice experiment and how the treatment profiles were developed.

The conclusions are consistent with the evidence and arguments presented in the paper, addressing the main question posed.

The references are appropriate, and the abstract correlates with the manuscript content.

Specific comment regarding lacks of the paper:

The paper could benefit from providing more detailed information about the development of treatment profiles and the specific design of the discrete-choice experiment.

Future perspectives to the authors:

The authors could consider conducting further research to explore pet owners' preferences for other treatment options in companion animal medicine, which would contribute to a more comprehensive understanding of this area.

Please double-check the reference list to ensure that all references are included in the main text and vice versa.

Suggestion for improvements:

Introduction:

Expand the introductory section to include a brief overview of the importance of safe pet feeding practices and food bowl hygiene in minimizing the risk of microbiological contaminations in the domestic environment.

Provide context for the comparative study you conducted on pet owners' perceptions of different formulations of oclacitinib by discussing the broader context of pet care practices, including feeding habits and bowl hygiene.

Discussion:

Discuss the findings of your study in light of the broader context of pet care practices outlined in the additional research. Compare and contrast the preferences and behaviors of pet owners regarding medication formulations with their practices related to feeding frequency, bowl material, and cleaning methods.

Highlight the potential implications of your findings on pet owners' preferences for chewable oclacitinib formulations in the context of broader pet care practices. For example, if pet owners are more inclined towards convenient and easy-to-administer medications, they might also prioritize convenience and efficiency in other aspects of pet care, such as feeding practices and bowl hygiene.

Implications and Recommendations:

Draw practical implications from both your study and the additional research on safe pet feeding practices and hygiene measures. Discuss the need for comprehensive guidelines and educational initiatives to promote safe feeding practices and proper food bowl hygiene among pet owners.

Provide recommendations for pet owners based on the combined findings of your study and the additional research, emphasizing the importance of regular cleaning of food bowls, appropriate selection of bowl materials, and adherence to safe food handling practices to minimize the risk of microbiological contamination in the domestic environment.

Future Research Directions:

Suggest potential avenues for future research based on the gaps identified in the existing literature and the implications of your study. For instance, future research could explore the effectiveness of different cleaning methods and materials in reducing microbiological contamination of pet food bowls across diverse pet owner demographics and geographical regions.

By integrating the information on safe pet feeding practices and food bowl hygiene into your paper, you can provide a more comprehensive understanding of pet owners' behaviors and preferences in caring for their pets and offer valuable insights for promoting pet health and well-being. I suggest read https://doi.org/10.1186/s12917-023-03823-w to find more information.

Statistical analysis:

I kindly request that you elaborate on the following aspects of the statistical analysis:

Description of the discrete-choice experiment methodology, including the specific design parameters and procedures used to compare the two formulations of oclacitinib.

Details on the statistical tests employed to analyze the survey data, including any regression models, hypothesis tests, or other statistical techniques used to assess the preferences of pet owners and determine significant differences between groups.

Explanation of how the data were processed and analyzed to derive key findings and conclusions regarding pet owners' preferences for chewable oclacitinib.

Providing additional information on these aspects of the statistical analysis would not only enhance the reproducibility of your study but also facilitate a deeper understanding of the methodologies employed and the validity of your research findings.

Dissemination:

As we reflect on the significance of your findings, we believe that adding a section on future perspectives would enhance the impact and practical applications of your research. Specifically, we suggest considering the following aspects for inclusion:

Dissemination via Social Media (see: 10.3390/ani13223503):

In light of recent studies that have demonstrated the effectiveness of social media platforms in disseminating research findings to a wider audience, we encourage you to discuss strategies for leveraging social media channels to reach pet owners, veterinarians, and other stakeholders interested in companion animal medicine.

Consider outlining specific approaches, such as creating infographics, short video summaries, or interactive content, to effectively communicate key findings and engage audiences across various social media platforms.

Integration into Veterinary and Animal Science Education (see: 10.1016/j.jevs.2023.104537):

Given the relevance of your research findings to veterinary practice and animal science, we recommend exploring innovative teaching methods to integrate your study into educational curricula.

Discuss the potential for incorporating your research findings into veterinary and animal science courses through case studies, practical demonstrations, or interactive workshops. Consider how these innovative teaching methods can enhance student learning outcomes and foster critical thinking about pet care practices and treatment options for canine pruritus.

By addressing these future perspectives, your paper can contribute to the broader dissemination of knowledge in the field of companion animal medicine and facilitate the translation of research findings into tangible educational resources and clinical practices.

Round 2

Reviewer 2 Report

Comments and Suggestions for Authors

the paper improved a lot, it is suitable for pubblication